# Introducing a Fair Tax Method to Harden Industrial Blockchain Applications against Network Attacks: A Game Theory Approach

**Fatemeh Stodt** *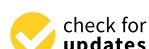 **and Christoph Reich** *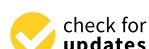

Institute for Data Science, Cloud Computing and IT Security, Furtwangen University of Applied Sciences, 78120 Furtwangen im Schwarzwald, Germany
* Correspondence: fatemeh.stodt@hs-furtwangen.de (F.S.); christoph.reich@hs-furtwangen.de (C.R.)

**Abstract:** Industrial Internet of Things (IIoT) systems are enhancing the delivery of services and boosting productivity in a wide array of industries, from manufacturing to healthcare. However, IIoT devices are susceptible to cyber-threats such as the leaking of important information, products becoming compromised, and damage to industrial controls. Recently, blockchain technology has been used to increase the trust between stakeholders collaborating in the supply chain in order to preserve privacy, ensure the provenance of material, provide machine-led maintenance, etc. In all cases, such industrial blockchains establish a novel foundation of trust for business transactions which could potentially streamline and expedite economic processes to a significant extent. This paper presents an examination of "Schloss", an industrial blockchain system architecture designed for multi-factory environments. It proposes an innovative solution to increase trust in industrial networks by incorporating a fairness concept as a subsystem of an industrial blockchain. The proposed mechanism leverages the concept of taxes imposed on blockchain nodes to enforce ethical conduct and discipline among participants. In this paper, we propose a game theory-based mechanism to address security and trust difficulties in industrial networks. The mechanism, inspired by the ultimatum game, progressively punishes malicious actors to increase the cost of fraud, improve the compensation system, and utilise the reward reporting capabilities of blockchain technology to further discourage fraudulent activities. Furthermore, the blockchain's incentive structure is utilised to reduce collusion and speed up the process of reaching equilibrium, thereby promoting a secure and trustworthy environment for industrial collaboration. The objective of this paper is to address lack of trust among industrial partners and introduce a solution that brings security and trust to the forefront of industrial blockchain applications.

**Keywords:** blockchain; Industry 4.0; IIoT; trust management; game theory; tax

## 1. Introduction

The rise of digitisation in industrial production environments as part of Industry 4.0 and the Industrial Internet of Things (IIoT) has led to the development of various modern use cases. These use cases include distributed production, external machine condition monitoring, machine maintenance by third parties, and spare part delivery on demand [1]. However, in order to ensure smooth operations it is crucial for all components, operators, managers, integrators, and stakeholders to work together seamlessly. Moreover, different situations require different levels of security and prioritisation within these systems.

Artificial intelligence (AI) is transforming the cybersecurity landscape by enhancing the capabilities of blockchain management systems and process mining approaches [2]. With the increasing complexity and volume of cyberthreats, blockchain-based security solutions are becoming more prevalent [3]. AI-powered blockchain management systems can detect and prevent cyberattacks by analysing patterns in data and identifying potential threats [4]. Additionally, the integration of blockchain in Industry 4.0 applications has

enabled process mining approaches that leverage AI to enhance cybersecurity [5]. By analysing transactional data on the blockchain, AI-powered process mining can detect anomalies and prevent fraudulent activity, thereby strengthening the security of industrial systems [6]. Overall, the combination of AI and blockchain is a promising approach to improving cybersecurity in various industries [7].

Maintaining the reliability and security of IIoT networks is essential for long-term success. Blockchain technology has emerged as a key tool in managing the risks associated with these networks. By combining cybersecurity measures, assurance services, and best practices, blockchain networks can minimise the risk of attacks and fraud [8]. However, it is important to note that security in a blockchain network depends on the behaviour of system participants, not just the technology itself [9]. Although traditional industrial networks such as Modbus, Profinet, KNX, CanBus, BacNet, and EtherCat remain widely used in Industry 4.0 and are unlikely to be replaced entirely by blockchain-based networks, integrating blockchain technology with these networks through gateways or adapters can provide secure and scalable data exchange while maintaining the functionality of existing networks.

In the context of IIoT, blockchain technology provides a decentralised and secure infrastructure for data exchange, enabling secure and efficient machine-to-machine communication. It can be used to incentivise positive behaviour among system participants and establish a fair taxation system for resource distribution. Therefore, the combination of IIoT and blockchain technology offers a promising solution for enhancing the security, scalability, and efficiency of industrial production environments.

To regulate the behaviour of participants in an industrial blockchain network, a non-cash-incentive taxation system can be implemented to fund a reward system that incentivises positive behaviour and penalises immoral behaviour. In this sense, blockchain networks can be seen as distributed networks similar to human societies, especially when considering the growing interconnectedness of IoT devices and the expanding networks they are a part of.

Fairness in computer networks deals with the allocation of resources across applications; fairness is achieved when resources are distributed equitably [10]. The investigation of fairness in computer networks serves two purposes: first, to improve network behaviour by promoting equitable resource distribution, and second, to enable new fair applications that are currently not feasible in existing networks. The distributed nature of blockchain networks can be compared to social behaviour [11]; fairness can be established in the network through a taxation system. The aim of this paper is to introduce a **fair taxation system** in a blockchain network as a subsystem of the Schloss system architecture [12]. The Schloss system architecture seeks to establish collaboration between industrial networks in a distributed manner. This project introduces a distributed blockchain-based architecture for IIoT that incorporates a measuring system to evaluate the behaviour and trustworthiness of nodes in the network, thereby creating a trustworthy environment within which different participants in industrial processes can collaborate.

The key contributions of this paper are:

- A taxation scheme for blockchain participants to deter malicious attacks.
- The introduction of a game theory mechanism within the network to establish fairness among nodes.

The rest of this paper is structured as follows: Section 2 reviews related works in the field; Section 3 outlines the system issues and objectives; Section 4 explains the schema design; Section 5 addresses security concerns; and Section 6 provides the conclusions.

## 2. State of the Art

The issue of fairness in the rapidly developing world of new technologies, especially with the growth of AI and IoT, has sparked many discussions and debates. In their research, Huelsen et al. highlighted the vulnerabilities in social networks, IoT, blockchain, and AI,

and stressed the importance of considering fairness problems associated with these new technologies [13].

As a result, researchers have been searching for solutions to address fairness in networks. Li et al. proposed an IoT security threat model that uses deep learning to create a fair environment by autonomously learning from attacks [14]. Additionally, Hu et al. discussed the vulnerability of trust relationships between network nodes and introduced a blockchain-based software-defined network architecture that rewards nodes based on contract theory to achieve fairness [15].

Fairness is a critical aspect of network design and operations; it entails ensuring that all nodes have an equal opportunity to perform tasks based on their computational capacity. Wang et al. proposed an information exchange protocol for IoT devices that uses entropy measurement in rational information exchange protocols to promote fairness [16]. The authors argued that this protocol can enable IoT devices to interact fairly, making it possible to achieve a higher level of efficiency and security.

In the healthcare industry, fairness is crucial to the provision of quality services. Nwebonyi et al. proposed a protocol for ensuring fairness and security in IoT-driven healthcare by employing a blockchain-based approach [17]. The authors suggested that blockchain technology could be used to ensure that all stakeholders in the healthcare industry have equal access to data while at the same time preserving privacy and security.

Efficient management of industrial IoT data is another critical area where fairness is required. Jiang et al. suggested a heuristic and min-heap-based optimal algorithm for putting industrial IoT data into permissioned blockchains [18]. The proposed algorithm is designed to ensure that all nodes have equal access to data and computational resources. By promoting fairness in the distribution of data, the authors argued that this algorithm can contribute to the overall efficiency of the system.

IoT devices are designed to maximise their value while ensuring security within the constraints of their hardware and software. One approach to understanding the behaviour of these devices is to view them through the lens of game theory, which considers the strategic interactions of rational actors in situations of conflict or cooperation [19]. By modelling the behaviour of IoT devices as rational, game theory can provide insights into their decision-making processes and help identify strategies for achieving desirable outcomes.

However, the use of game theory in the context of IoT networks is not without challenges. For instance, a key problem is preventing cheating and fraud in the process of issuing transactions [20]. Researchers have proposed various approaches to address this issue, including the use of smart contracts and consensus mechanisms to ensure the integrity of transactions [21].

Game theory has been applied to discussions on taxation as well, with a focus on preventing tax evasion. Research in this area has explored various strategies for incentivising compliance and discouraging fraud, such as increasing penalties for non-compliance or introducing rewards for good behaviour [22,23]. By leveraging game theory and related tools, researchers can gain a deeper understanding of the challenges involved in securing IoT networks and develop more effective solutions to address these challenges.

Blockchain technology has gained widespread attention for its potential to increase the efficiency of asset management and to reduce transaction costs in the commercial banking sector [24]. Beyond the financial industry, numerous studies have explored the potential applications of blockchain technology to improve efficiency and fairness in other areas, including taxation. For instance, Sogaard et al. investigated the use of blockchains to reduce administrative burden and enhance transparency in tax administration [25]. Alzubi et al. proposed a modified blockchain-based tax compliance framework that integrates smart contracts and machine learning to achieve fairness and transparency [26]. Similarly, Yayman et al. proposed a blockchain-based solution to enhance tax compliance and reduce fraud in the tourism industry [27]. Wang et al. explored the application of blockchain technology in the field of e-government and proposed an architecture for secure and transparent tax collection and management [28]. In addition, Ayyappath et al. presented a

design framework for a blockchain-based tax compliance system that incorporates game theory and smart contracts to promote fairness and transparency [29]. Finally, Pelaez et al. investigated the potential of blockchain technology to improve the fairness and efficiency of the tax collection process by enhancing transparency and reducing corruption [30].

However, despite the focus on using blockchain for taxation purposes, no one has previously proposed using taxation-inspired methods as part of a blockchain network to achieve greater fairness for participants.

## 3. Attack Vectors and Challenges for Blockchain Applications

### 3.1. Security Risks

The security of blockchain technology is often praised as being very strong; however, it is directly tied to the amount of hash power supporting the network. In this section, we discuss the most common attack scenarios against blockchain systems. The issue with blockchains is that majority power can have a negative impact. This is because the blockchain operates on a consensus system in which all decisions are made by a majority of the nodes. Unfortunately, this means that nodes with a majority of computational power or stake can take advantage of this to make decisions that suit their own interests.

#### 3.1.1. Sybil Attack

The Sybil attack scenario involves three types of nodes: honest, Sybil, and attacker nodes, as depicted in Figure 1. To carry out the attack, the attacker creates multiple Sybil nodes and connects them to honest nodes, thereby disrupting the authentic connections between honest nodes in the P2P network. When the attacker acquires a disproportionate level of control over the P2P network, they can take control of it. The attacker can then use their Sybil nodes and an attack method to launch various threats that harm the reputation system of the P2P network.

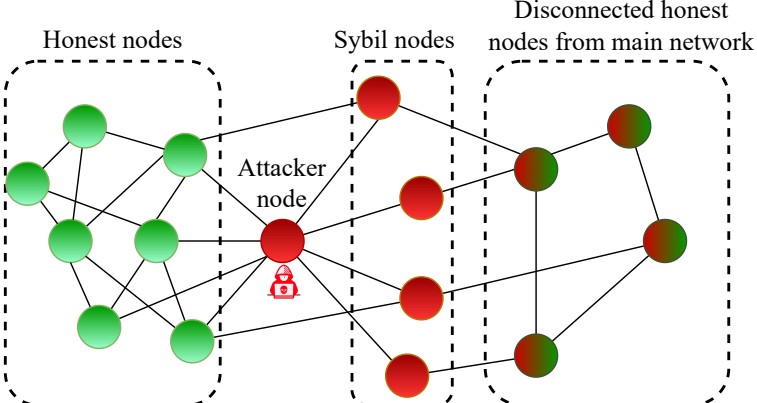

**Figure 1.** Sybil attack scenario in a P2P network.

In a distributed network, an attacker can carry out two different types of Sybil attacks, known as "Sybil community" and "Scattered Sybil".

In the case of a "Sybil community" attack, as illustrated in Figure 2a, the attacker creates many connections within the Sybil community, meaning that Sybil nodes are strongly connected to each other. However, the Sybil community has limited ability to connect with trustworthy nodes. In other words, there are only a few connections between Sybil nodes and honest nodes.

**Industrial Blockchain Issue 1:** In industrial networks or blockchain, attackers may try to introduce a Sybil community as a new sub-network or small company in the industrial network in order to launch a Sybil attack against the industrial blockchain.

A "Scattered Sybil" attack involves attackers present in multiple domains, as shown in Figure 2b. Unlike a Sybil community attack, a scattered Sybil attack is able to form connections with regular users in addition to those with other Sybil identities. This ability

to imitate the social structure of a typical user makes scattered Sybil attacks particularly dangerous, as attackers can gain access to many honest nodes.

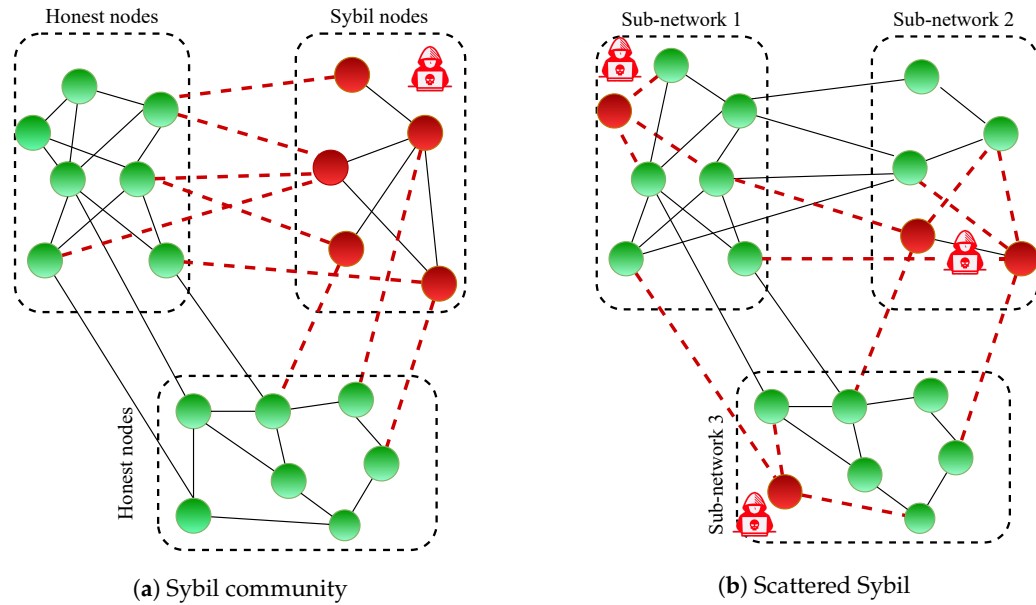

(**a**) Sybil community      (**b**) Scattered Sybil

**Figure 2.** Sybil attack.

**Industrial Blockchain Issue 2:** In industrial networks or blockchains, attackers may attempt to spread out their Sybil nodes across different sub-networks or organisations in order to gain varying access to honest nodes and eventually infect the entire network with Sybil nodes.

### 3.1.2. Double-Spend Attack

In a double-spend attack, an attacker attempts to create a separate ledger version by forking the original blockchain (Figure 3a). Blocks are then generated on this new ledger without sharing it with the public blockchain network. This results in two different ledgers, one controlled by the attacker and the other by the honest nodes.

When the attacker pays for a product, they transfer the money to the seller. However, they only spend the money on the honest nodes' ledger, not on their secret ledger. The network's nodes always consider the longest chain to be the valid one (Figure 3b). If the attacker gains more computational power and builds a longer chain faster, the honest nodes will eventually accept the attacker's chain as the valid one.

As a result, the transaction accounting for the money is no longer on the chain, while the attacker has already received the merchandise, leaving the seller without payment. The attacker is then free to use the money for other purposes.

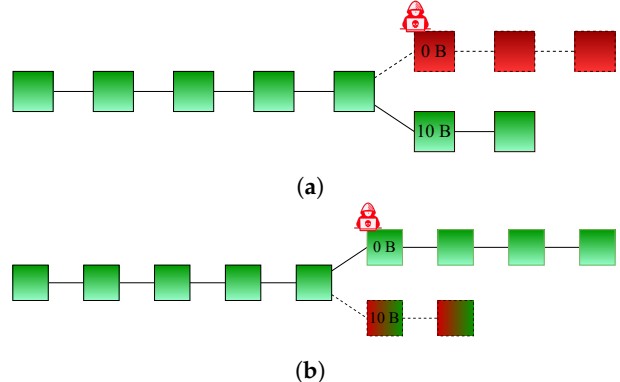

(**a**)

(**b**)

**Figure 3.** Ledger state in a Double-Spend attack. (**a**) Create a separate ledger version by forking the original blockchain by attacker. (**b**) Takeover the main chain by attacker.

### 3.1.3. 51% Attack

The 51% attack is the most lethal hash rate-based attack on a blockchain [31]. As an example, numerous alternative cryptocurrencies (alt-coins) with vastly different market capitalisations have been introduced. This has made 51% assaults against alt-coins plausible, as only a small percentage of miners from larger currencies must transfer to a smaller currency in order to control 51% of the network hash rate of the smaller coin. This has led to the development of economic models that take into account the motivations for launching a 51% assault in a world where a sufficient hash-rate can be obtained if the attacker is prepared to pay [32]. These theories imply that successful attacks must be either lucrative or break even unless miners have significant fixed expenses connected with their mining gear that cannot be recovered in the event of an attack.

Table 1 shows a summary of related attacks on blockchain networks and analyses the nodes' behaviour in the different attack cases. By studying the behaviour of the nodes, it is possible to consider their specific behaviours, which can provide useful hints for designing a mechanism to prevent various attack types in the proposed Schloss system.

**Table 1.** Behavioural Comparison in Related Attacks.

| Attack Name | Features | Attack Goal | Nodes Behaviour | Related Issue |
|---|---|---|---|---|
| Sybil Community | There are malicious nodes in the same sub-network or community, and the number of attack linkages to honest nodes is restricted. | Intentionally upload malicious transactions or wrong voting to manipulate the entire system. | Behave as normal nodes and regularly repeat certain behaviours. | Issue 1 |
| Scattered Sybil | Malicious nodes are spread in each sub-network, and the number of attack linkages to honest nodes is higher. | Distribute spam and malware in order to launch other attacks, or invade the privacy of other users. | Intentionally repeat a certain behaviour often. | Issue 2 |
| Double-Spend | The hidden fork must be mined faster than the original blockchain. | The danger that a cryptocurrency may be used twice or more is known as double-spending. | No activity or updates from nodes for quite some time. | Issue 3 |
| 51% Attack | A group of miner nodes that control over 50% of the network's mining hash rate or stake. | Prohibit other miners from publishing blocks if they control the majority of the power on the network. | According to Ethereum statistics, the longest chain of mined blocks by the same mining pool equates to 17 blocks, which happens with a frequency of 0.000008% over a given time [33]. | Issue 3 |

**Industrial Blockchain Issue 3:** Considering an industrial blockchain network, it is necessary to keep newly joined accounts and miners under supervision in order to prevent a 51% attack. Due to the smaller size of industrial blockchain networks, it is easier to obtain a majority; therefore, such networks are even more vulnerable to this type of attack.

### 3.2. Inflation

Incentive mechanisms represent a solution that can ensure nodes' good behaviour. The idea of providing rewards to the most trustworthy nodes, it is possible to encourage them to maintain good behaviour. Problems may arise, however, when the blockchain needs to provide the nodes with their reward.

Consider a blockchain network in which payment of currencies to nodes is required for each mining block or operation and there is no limit to the quantity of coins in the network. This results in network inflation [34], meaning that the blockchain loses its value; thus, it is forced to issue more coins to complete tasks, further depreciating the coins' worth. For this reason, a new kind of digital money called a *stablecoin* is gaining popularity. However, if a stable-coin is linked to a particular fiat currency it is subject to inflation in the reserve currency, and may ultimately lose value as the this currency depreciates. Thus, in order to govern a blockchain network there must be a finite currency supply and a method of circulating money inside the blockchain without inducing inflation.

**Industrial Blockchain Issue 4:** In the case of industrial networks, the production of an infinite number of coins via the industrial blockchain would result in inflation. If the fairness is merely determined by the amount of money and all spending is on node transactions, the wealthiest firms always enjoy the majority of network power.

## 4. Fairness Solution

To understand the security vulnerabilities and issues in a network, the impact of malicious node behaviour can be evaluated by examining Tables 1 and 2. The behaviour of nodes in a network can reveal their underlying intentions. To ensure that the network is trustworthy, it is essential to create a strategy and mechanism design that incentivises good behaviour among nodes. Mechanism design theory involves designing mechanisms that produce desired outcomes and takes into account the individual incentives and motivations of nodes. To encourage trustworthy behaviour, a fairness-based taxation approach can be applied to network nodes. This creates an incentive mechanism by making it advantageous for nodes to act in a trustworthy manner. Game theory can be used to determine the best strategy for each type of node and develop the mechanism to achieve the overall objective of a secure network.

**Table 2.** Overview of the issues.

| Industrial Blockchain Issue | Effect on System |
| --- | --- |
| Issue 1 | Scattered Sybil attacks involving attackers present in multiple industrial sub-networks |
| Issue 2 | Double-Spend attacks |
| Issue 3 | 51% attacks and majority attacks |
| Issue 4 | Inflation in case of new coins created for rewards |

### 4.1. Methodology of Game Theory: Main Features and Pillars

Game theory is a mathematical framework used to model strategic interactions between rational decision-makers. The main features of game theory include the use of formal models to represent interactions, the assumption of rationality on the part of players, and the analysis of outcomes in terms of payoffs or utilities. The following pillars form the foundation of game theory:

Players: Game theory assumes that there are at least two players involved in a strategic interaction. These players can be individuals, groups, or even countries.

Actions: Players have a set of possible actions available to them, and they choose an action based on their preferences and beliefs about the other players' actions.

Payoffs: The outcome of a game is measured in terms of payoffs or utilities. These payoffs represent the benefits or costs to each player from the various possible outcomes.

Information: Players may have different levels of information about the game and the other players. Game theory distinguishes between complete information games in which all players know all aspects of the game, and incomplete information games in which certain aspects of the game are unknown to certain players.

Strategies: Players can adopt different strategies based on their preferences and beliefs. A strategy is a plan of action that a player chooses in order to achieve their objectives.

Equilibrium: A solution to a game is an equilibrium, where each player's strategy is a best response to the strategies of the other players. The most commonly studied type of equilibrium in game theory is the Nash equilibrium, where no player has an incentive to deviate from their strategy given the strategies of the other players.

In summary, game theory provides a powerful tool for analysing strategic interactions between rational decisionmakers. By modelling interactions between players, their possible actions, and the outcomes of those actions, game theory allows us to understand and predict behaviour in a wide range of settings, from economics and politics to social interactions and evolutionary biology

*4.2. Definition of the Game*

A system for fair taxation establishes the foundation for an ultimatum bargaining game [35] based on concerns about the network's goals and the fairness of charging network nodes. The ultimatum game (Figure 4) has grown in popularity as a tool for economic experimentation. The ultimate game can be modelled as follows:

**Actors:** Each node in the network can be considered as an actor.

**Strategy:** Each node has two strategies, i.e., to be either trustworthy or malicious.

**Payoffs:** The payoffs for each node can be modelled as follows:

1. If a node is trustworthy, it receives a reward in the form of a fraction of the tax collected from all the nodes in the network.
2. If a node is malicious, it incurs a punishment in the form of the blocking of its deposit.

**Game Formulation:** The game can be formulated as a two-player non-cooperative game where each node is a player and its strategy is to either be trustworthy or malicious. The payouts for each player depend on the strategies of all the players in the network.

**Game Solution:** The Nash equilibrium, where each node chooses its strategy such that it cannot improve its payoffs by changing its strategy unilaterally, is the solution to the game.

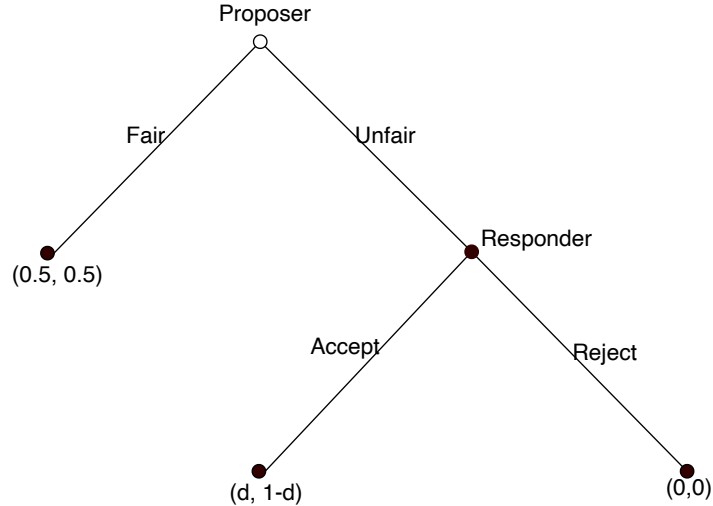

**Figure 4.** Extensive version of the mini-ultimate game.

Here, we explore a straightforward scenario with two players (notation and definitions are defined in Table 3), the full node of the blockchain as Proposer ($P$) and the light node in the blockchain as Responder ($R$). Considering a tax relationship in which $P$ wishes to suggest a tax for an item from $R$ and the profit ($PF$) created by the deal is more than zero ($PF > 0$), we suppose that there are two feasible outcomes and that each player has a preference for one of them. Let $PF_\theta \in [0, PF]$ and $\theta \in \{P, R\}$ denote the trade profit achieved by Player $P$ in outcome $\theta$, and Player $R$ denote the excess realised by Player $P$ in outcome $\theta$. We assume, without sacrificing generality, that $PF_P > PF_R$. In this example, Player $P$ prefers result $P$, while Player $R$ prefers result $R$. For the sake of clarity, we suppose that they barter over the likelihood of each player being picked to execute his or her proposed conclusion. The game's timing is as follows:

1. Player $P$ distributes the probabilities of achieving the preferred outcome ($\theta_{prefered}$) by selecting $\theta_{prefered}[0, 1]$; the probability that Player $P$ will implement their own desired result is ($1 - \theta_{prefered}$), with the probability that Player $R$ will implement $\theta_{prefered}$ being their ideal result.
2. Player $R$ evaluates their probability of achieving the desired result and may accept or reject it.

3.  The ambiguity is resolved, with payoffs $(x, y) - x$ being realised for Player $P$ and payoffs $y$ being realised for Player $R$. If Player $R$ accepts and Player $P$ is able to achieve the desired result, the payoffs are $(PF_P, (PF - PF_P))$. If Player $R$ accepts and is able to carry out the selected result, the payoffs are $(PF_R, (PF - PF_R))$. If Player $R$ is rejected, there is no trade, and the payoffs are be $(0, 0)$.

**Table 3.** Notation and definitions.

| Symbol | Definition |
|---|---|
| PF | Profit |
| $\theta$ | Outcome |
| $\theta_{prefered}$ | Prefer outcome |
| $I_{envy}$ | Aversion to disadvantageous inequality |
| $I_{guil}$ | Aversion to advantageous inequality |

In Step 1, Player $P$ has the complete opportunity to implement their own preferred outcome. In Step 2, standard game-theoretic reasoning predicts that Player $R$ will take any opportunity to achieve their own chosen result. Below is the utility function based on the fairness model of Fehr and Schmit [36]. The utility function is as follows:

$$\tilde{u}(x, y) = x - I_{envy}.max[0, y - x] - I_{guil}.max[0, x - y] \tag{1}$$

which assumes that $I_{envy} \geq I_{guil} \geq 0$ and $I_{guil} \leq 1$.

Considering risk aversion utility function $v(x)$, where $r$ is the risk tolerance parameter, we have

$$\tilde{u}(x, y) = v(x) + x - I_{envy}.max[0, y - x] - I_{guil}.max[0, x - y] \tag{2}$$

where $r < 1$ denotes risk aversion and $r > 1$ denotes risk-preferring tendencies. The overall risk preferences do not exhibit constant relative risk aversion unless $I_{guil} = 1$, meaning that we are in the domain of advantageous inequality. Based on [37] for all $I_{guil} \in [0, 1]$ and 2, the fairness of the expected outcome is equal to

$$\Omega(F) = u(E[x], E[y]). \tag{3}$$

Considering (3), the players assess the utility function's own-payoff component using anticipated values. Because a proposed allocation of $\theta_{prefered} = 1/2$ ensures expectation payoff equality, a rational Player $P$ never offers more than this to Player $R$ in equilibrium; therefore, it is better to focus on $\theta_{prefered} < 1/2$. In this case, the predicted inequality is detrimental to $R$.

Considering Satio's model [38], for anticipated inequality averse players, players assess their own reward in terms of expected utility; $R$ will accept the suggestion in this circumstance if and only if

$$u_r = \theta_{prefered}.PF^r + \theta_{prefered}.PF - I_{envy_R}.PF(1 - 2 * \theta_{prefered}) \geq 0. \tag{4}$$

The minimal level of approval for $R$ is

$$\theta_{prefered} \geq \frac{I_{envy_R}}{(1 + 2 * I_{envy_R} + PF^{r-1})}. \tag{5}$$

In this case, the utility of $p$ is:

$$u_p = (1 - \theta_{prefered}).PF^r + (1 - \theta_{prefered}).PF - I_{guil_p}.PF(1 - 2 * \theta_{prefered}) \geq 0. \tag{6}$$

The utility in (6) is positive even if $\theta_{prefered} = 0$ and $I_{guil_p} = 1$. Thus, the participants are more likely to reach an agreement, and on average the minimal acceptance thresholds for standard and hazardous ultimatum games should be identical.

Game theory plays a crucial role in the proposed mechanism. Each node in the network is viewed as an actor, and has two strategies, namely, to be either trustworthy or malicious. The payoffs for each node can be modelled as follows: if a node is trustworthy, it receives a reward R that is a fraction of the tax collected from all the nodes in the network. On the other hand, if a node is malicious it incurs a punishment P in the form of the blocking of its deposit.

The game is formulated as a two-player non-cooperative game, where each node is a player and its strategy is to either be trustworthy or malicious. The mathematical formulation of the game can be expressed as a matrix, where each row represents the payoffs for a node when it is trustworthy and each column represents the payoffs for a node when it is malicious. The Nash Equilibrium, where each node chooses its strategy such that it cannot improve its payoffs by changing its strategy unilaterally, is the solution to the game. In this case, no node has any incentive to deviate from its chosen strategy, as this could only result in a worse outcome.

### 4.3. Main Procedure: The Mini-Ultimatum Game

The proposed mechanism aims to address the issues of inflation and dominance of powerful nodes by incorporating a tax rate, which is calculated based on the amount of money in each light node's wallet. The network takes into account the possibility of punishment and makes it more difficult for malicious actors to target nodes.

The proposed procedure for our system is depicted in Figure 5 and comprises six steps that are performed for each epoch. These steps are carried out by full nodes that act as committee members responsible for running the incentive mechanism. In order to determine the fair amount of tax that each light node should pay, the committee members use game theory concepts that take into account the light node's behaviour and wallet balance within the network.

More specifically, the game theory approach employed in our system is designed to incentivise light nodes to behave in a way that benefits the overall network while ensuring fairness and transparency in the tax collection process. To achieve this goal, the committee members use a token-based mechanism (TMS) design that suggests the appropriate tax amount for each light node based on its performance in the network.

The TMS algorithm takes into account a range of factors, including the light node's transaction history, network activity, and overall contribution to the network's security and stability. This information is used to generate a score for each light node, which is then used to determine the amount of tax it should pay. Using this approach, the tax collection process is fair and transparent and the light nodes are incentivised to contribute positively to the network.

Algorithm 1, "Initialise", sets up the necessary parameters such as the certificate status, ledger, and wallet amounts for each participant. It checks whether the ledger is positive and whether each participant's wallet is equal to or greater than a certain amount; then, it sets the participants, subtracts a specific amount from the full node's wallet, and sets the deposit and reward amounts.

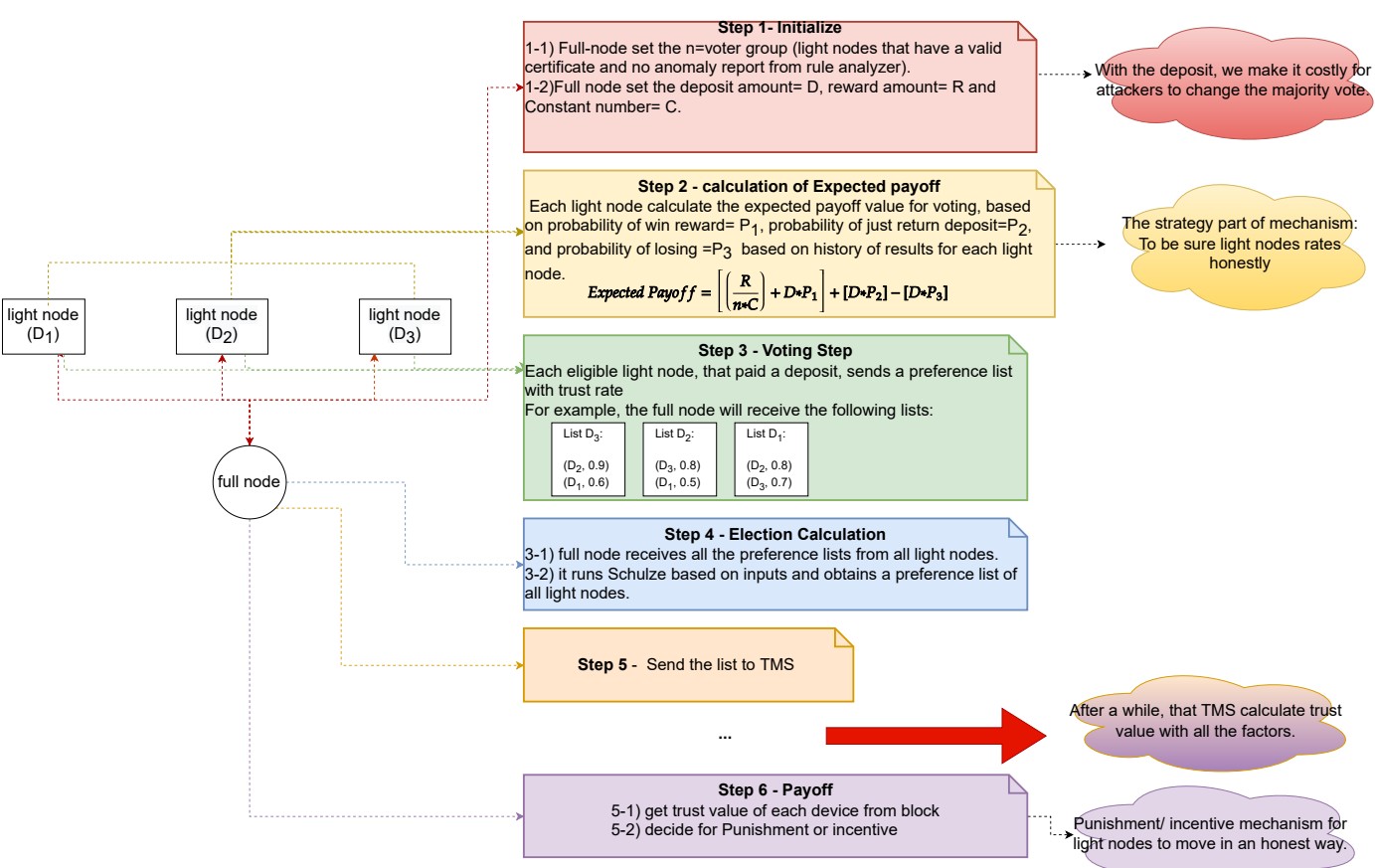

**Figure 5.** Extensive version of the mini-ultimate game.

---

**Algorithm 1** Initialise

---

**Input**: Eligible Nodes, Certificate Status, Rule Anomaly Status, Ledger, FullNode, LNi, BD, D, C

1: $state \leftarrow$ ''Init''
2: **if** $Ledger[FullNode] < 0$ **then**
3:     **return** False
4: **end if**
5: **for** $i \in LNi$ **do**
6:     **if** $wallet[i] < D$ **then**
7:         **return** False
8:     **end if**
9: **end for**
10: $participants \leftarrow LNi$
11: $CandidateGroup \leftarrow LNi$
12: $wallet[FullNode] \leftarrow wallet[FullNode] - \Sigma R$
13: $deposit \leftarrow D$
14: $reward \leftarrow R$
15: $constant \leftarrow C$
16: **for** $i \in LNi$ **do**
17:     $wallet[i] \leftarrow wallet[i] - D$
18: **end for**
19: $state \leftarrow$ ''PayoffCalculation''
20: **return** True

---

Algorithm 2, "Expected Payoff Calculation", calculates the expected payoff for each participant by adding and subtracting specific amounts and dividing by the number of participants. The result is stored as "ExpectedPayoff".

---

**Algorithm 2** Expected Payoff Calculation

---

**Input**: Participants, Reward, Constant, Deposit, P1, P2, P3

1: Initialise:
2:     a. Set p1, p2, and p3.
3:     b. Calculate the expected payoff:
4:         expected payoff = $\left( \frac{Reward}{(\text{number of participants} \cdot Constant)} + Deposit \cdot p1 \right) + (Deposit \cdot P2) - (Deposit \cdot p3)$
5:     c. Store the result in a variable "ExpectedPayoff".

---

Algorithm 3 is the main algorithm; it takes inputs from the other two algorithms and other sources and outputs the "Trusty Nodes" and "Rewards". It starts by setting the reward and deposit amounts, running the Tax algorithm to announce the tax amount to the light nodes, and applying send transactions for each light node, then calculates the expected payoff and sends the deposit to the blockchain.

The algorithm collects the preference lists of all nodes and runs the Schulze algorithm to determine the trust value of each node. The top 10% of trustworthy nodes are selected and rewarded, malicious nodes with low trust values have their deposits blocked, and trustworthy nodes have their deposits returned.

---

**Algorithm 3** Algorithm for Incentivising Good Behaviour in the Network

---

**Input**: Epochs, Initialise algorithm, Tax algorithm, Expected Payoff algorithm, Schulze algorithm, TMS

**Output**: Trusty Nodes, Rewards

1: **for** each epoch **do**
2:     Get input from Initialise algorithm
3:     Set the reward as R, deposit as D, and constant number as C from full nodes to the network
4:     Run Tax algorithm and announce tax amount to light nodes
5:     **for** each light node per epoch **do**
6:         **while** wallet amount > tax amount **do**
7:             Apply send transaction
8:         **end while**
9:         Run Expected Payoff algorithm and send deposit to blockchain
10:     **end for**
11:     **for** each node that paid deposit **do**
12:         Join the vote to the neighbours and create a preference list
13:     **end for**
14:     **for** each local network full node **do**
15:         Receive all the preference lists from light nodes and run Schulze algorithm to obtain a preference list of all light nodes
16:     **end for**
17:     **for** all full nodes in the network **do**
18:         Collect the preference lists of all sub-networks as input for TMS
19:     **end for**
20:     Calculate the trust value for each node in the network using TMS
21:     **for** each node in the network **do**
22:         **if** node is malicious and has low trust value **then**
23:             Block deposit
24:         **else if** node is trustworthy **then**
25:             Return deposit
26:         **end if**
27:     **end for**
28:     Select the top 10% trusty nodes and pay reward to them
29: **end for**

---

*4.4. Real-World Use Case*

Consider a real-world scenario where three stakeholders, A, B, and C, are part of a decentralised industrial factory collaboration. A, B, and C are expected to behave well; however, bad behaviour may nonetheless occur. Algorithm 1 sets up the necessary parameters for each participant, such as their wallet amounts and the deposit and reward amounts. Algorithm 2 calculates the expected payout for each participant based on the inputs received from Algorithm 1. Algorithm 3 takes the inputs from the other algorithms and runs the necessary computations to determine the trust value of each node. Initially, all three nodes are selected as trustworthy, and are rewarded accordingly.

However, after a few rounds of interactions, stakeholder B begins to provide false inputs in an attempt to cheat the other stakeholders. This results in the trust value of stakeholder B being reduced, and they are penalised accordingly. Over time, if stakeholder B continues to engage in bad behaviour their trust value could decrease even further, potentially resulting in their removal from the network.

Meanwhile, stakeholders A and C continue to behave well and provide accurate inputs, resulting in their trust values increasing over time. As a result, they may receive more responsibility within the network and potentially even be rewarded with additional benefits.

This demonstrates the importance of having a trust algorithm that is able to adapt to changing circumstances and account for bad behaviour when it occurs while continuing to reward good behaviour.

*4.5. Advantages of Applying the Tax Game to Blockchains*

The purpose of this paper is to propose a tax mechanism based on game theory for enhancing trustworthiness in industrial blockchains. To the best of our knowledge, the incorporation of taxation into blockchains has not been previously explored. The tax mechanism proposed in this paper operates on a per-account basis and applies to all node activities, including those from malicious entities that enlist in the blockchain using fake identities. The primary goals of this approach are to ensure fairness in node activities and to prevent majority attacks.

The design of a blockchain tax should be guided by the following objectives:

1.  Decentralisation: as the number of nodes in the network increases, centralised trust management techniques may become infeasible. This strategy, informed by the trust management system described in Schloss, employs a decentralised approach that leverages full nodes and light nodes to distribute trust values and ensure system stability and scalability.
2.  Network fairness and balance: using trust values to assess nodes and impose taxes, this mechanism helps to maintain honest nodes in the network. The combination of being honest and facing an uncertain time limit presents a challenge for malicious nodes, as it is contrary to their objective and causes harm to honest nodes.
3.  Deterring and complicating the work of hackers and malicious agents: enrolling in and paying taxes for an indefinite period of time while accounting for the production of additional nodes for a majority attack is cost-prohibitive. Additionally, as participants in this game are assumed to be rational, remaining in the network as a malevolent entity is not advantageous or cost-effective.

**5. Discussion and Security Analysis**

This section discusses security aspects of the suggested technique.

***Industrial Blockchain Issue Privacy:*** The use of private and public keys is a critical component of blockchain privacy. To secure user transactions, blockchain systems use asymmetric cryptography. In these systems, each user has a public and a private key. These keys are cryptographically-linked random sequences of numbers. It is theoretically impossible for a user to determine another user's private key from their public key. This makes hostile nodes unable to track an overlay node.

*Security:* The blockchain is primarily responsible for the security of the design. Each transaction in the blockchain is accompanied by a data hash, which protects the integrity of the data. All transactions are encrypted using asymmetric encryption algorithms that guarantee privacy. Consequently, the issues mentioned in Section 3 around an attacker being able to seize control of nodes or the network addresses are addressed:

*Addressing Industrial Blockchain Issue 1*: Addressing the addition of a new small company or workshop with new devices, the new stakeholder must obtain the consent of the private blockchain's members and other stakeholders. In this situation, the danger of establishing a Sybil community is too low, as members are unwilling to add an unknown company. In the Schloss architecture, each node is rated based on its neighbours with known trustworthy nodes. The network is then divided into two divisions depending on the tax method's test preference. These settings define the partition's border, often known as the "cutoff". Hence, nodes that are firmly linked to honest reputable ones are ranked higher.

*Addressing Industrial Blockchain Issue 2*: The purpose of a distributed Sybil attack is to spread spam, ads, and malware, steal and breach users' privacy, and intentionally influence the reputation system. Malicious nodes in the network masquerade as regular users while purposefully disseminating spam, ads, and viruses. In the proposed method for enrolling new devices, which takes into account the mechanism's design and the proposed technique, new devices must register for voting, however, their vote is not yet taken into account; instead, their behaviour is monitored for an undetermined number of iterations. The rogue nodes have little effect on the network, as in order to have an effect to the node's status it is necessary to have numerous positive rates in order to overstate the target node or many bad rates in order to undervalue it.

*Addressing Industrial Blockchain Issue 3*: In the case of a Sybil node, it is not advantageous for an attacker to place devices in the network for a long amount of time while they are behaving in an honest manner. Assuming that a malicious node will behave ethically, eventually allowing it to vote and have its votes counted, it must pay a deposit and tax. In the case of malicious voting, the deposit is stopped by the full nodes, and the malicious node is placed back into the test cycle. Therefore, it is too expensive for an attacker to maintain malicious nodes.

*Addressing Industrial Blockchain Issue 4*: The full nodes calculate the tax required for each node to be permitted to vote. This is contingent on the budget required for rewarding the most trustworthy nodes and the quantity of money in each node's wallet. In this situation, the blockchain prevents inflation and money circulation within the network. While the tax is based on the amount of money in each wallet, it is less than the award, prevents tax requests from being denied. Moreover, using a random integer $N$ to test newly enrolled devices prevents them from receiving network power. The network makes decisions about who can operate the system by selecting the nodes that prove most reliable over time. Table 4 shows a summary of related issues and then summarises the solutions offered by proposed method.

**Table 4.** Summary of solutions offered by the proposed fairness system.

| Issue | Solution |
|---|---|
| Issue 1 | Adding new members to the network requires consensus |
| Issue 2 | Newly enrolled devices have their behavior monitored for an undetermined number of iterations to minimize their effect on the network. |
| Issue 3 | 51% Requiring a deposit and monitoring the behavior of new nodes makes it expensive for attackers to maintain malicious nodes in the network. |
| Issue 4 | The full nodes calculate the tax for each node based on the amount of money in their wallet, preventing inflation and money circulation. |

## 6. Conclusions

In conclusion, this paper presents a novel decentralised taxation mechanism that incorporates a trust management system in an industrial network. By implementing the Schloss subsystem, full nodes are able to assess the credibility of transactions based on the trust values of all nodes stored on the blockchain. The tax is aggregated among the full nodes and distributed based on the trust values and wallet status of each node. This system provides a secure and reliable environment for all nodes to work together, and prevents majority attacks and inflation within the blockchain. The exact tax amount and the undisclosed iteration number ensure the integrity of the system. Our proposed mechanism has the potential to establish a safe and efficient intelligent industrial network, and provides an incentive structure that promotes fairness and reliability.

As a future research direction, we suggest investigating the potential role of sustainability and digital transition issues in further developing the proposed mechanism. Further research could explore how the proposed mechanism converges or diverges from other approaches. The theoretical justification can be strengthened by discussing how the proposed mechanism compares to previous methodological contributions. By addressing these areas, the proposed mechanism can be further validated, contributing to the development of more efficient and secure industrial networks.

**Author Contributions:** Conceptualization, F.S.; writing—original draft preparation, F.S.; review and editing, C.R. All authors have read and agreed to the published version of the manuscript.

**Funding:** This research was funded by the Federal Ministry of Education and Research (BMBF) under reference number COSMIC-X 02J21D144, and supervised by Projektträger Karlsruhe (PTKA).

**Data Availability Statement:** All data were presented in main text.

**Conflicts of Interest:** The authors declare no conflict of interest.

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
