# Peer review of "Introducing a Fair Tax Method to Harden Industrial Blockchain Applications against Network Attacks: A Game Theory Approach"

_computers, doi:10.3390/computers12030064_

Round 1

Reviewer 1 Report

The paper presents taxation system in blockchain network.

Do you think blockchain will be integrated into Industrial Networks, like: Modbus, Profinet, KNX, CanBus, BacNet, EtherCat? If I understand correctly, using Raspberry PIs as gateways, that decentralized network can be implemented, but not in Industrial Network. In Industry 4.0 Profinet, KNX etc. they will not disappear, but will expand using other devices as a gateways. This aspect did not comes out when reading the article. If you agree with me please add a few sentences about it and/or avoid using the term Industrial Network.

I have no other comments. I liked the work.

Author Response

We appreciate the time and effort that you have dedicated to providing your valuable feedback on our manuscript. We are grateful for the insightful comments on our paper. Furthermore, we have been able to incorporate changes to reflect most of the suggestions provided. We have highlighted the changes within the manuscript. Here is a point-by-point response to the comments and concerns.

Comments : Do you think blockchain will be integrated into Industrial Networks, like: Modbus, Profinet, KNX, CanBus, BacNet, EtherCat? If I understand correctly, using Raspberry PIs as gateways, that decentralized network can be implemented, but not in Industrial Network. In Industry 4.0 Profinet, KNX etc. they will not disappear, but will expand using other devices as a gateways. This aspect did not comes out when reading the article. If you agree with me please add a few sentences about it and/or avoid using the term Industrial Network.

Response: We appreciate your comment. We added some explanation regarding your comment in lines 47-52.

Reviewer 2 Report

In this paper, the authors deal with the problem of trust in industrial networks, especially in multi-factory environments with Industry 4.0 concept involved, where the Industrial Internet of Things (IIoT) systems/devices are more and more prevalent. Since recently, blockchain technology has been used to increase trust between stakeholders who can collaborate, for example, in the supply chain, distributed production, spare part delivery on demand, etc. Although in all those cases an industrial blockchain represents a platform of trust for business transactions, security in a blockchain network also depends on the behaviour of the system participants.

In the paper authors proposed a fair taxation system which regulates behaviour of the industrial blockchain network participants (full and light nodes) in a manner that positive behaviour is incentivised, and immoral behaviour is penalized. Thereby, an industrial blockchain system architecture (designed for multi-factory environments), named „Schloss“, is examined. Also, the Trust Management System (TMS), as a subsystem of Schloss (that can evaluate the behaviour and trustworthiness of nodes in the network), is used for provision of the trust values of participants/nodes, which are then used as input values for the algorithms incorporated in the taxation mechanism.

Beside the taxation scheme for blockchain participants (aimed to deter malicious attacks), the second key contribution of this paper is the introduction of a game theory mechanism (based on a version of the “mini-ultimatum” bargaining game) within the network to establish fairness among nodes. The proposed mechanism addresses the issues of inflation and of powerful nodes (51% attack), which are considered as two of four main industrial blockchain issues (besides scattered Sybil attack and Double-Spend attack).

The paper does not contain any simulations, neither provides any concrete results to support the proposed taxation scheme. Nevertheless, my opinion as the reviewer is that the theoretical contribution of the paper is significant and that the paper can be accepted.

Author Response

We appreciate the time and effort that you have dedicated to providing your valuable feedback on our manuscript. We are grateful for the insightful comments on our paper. Furthermore, we have been able to incorporate changes to reflect most of the suggestions provided. We have highlighted the changes within the manuscript regarding all comments.

Reviewer 3 Report

The paper “Introducing a Fair Tax Method to Harden Industrial Blockchain Applications Against Network Attacks: A Game Theory Approach” investigates an examination of an industrial blockchain system architecture designed for multi-factory environments and proposes an innovative solution to increase trust in industrial networks by incorporating a fairness concept as a subsystem of an industrial blockchain. To achieve this aim, it adopts a game theory approach. As for the theoretical background, my suggestion is to clarify how the literature on the topic was reviewed to justify the research framework.

The authors should improve the theoretical justification of the study clarifying the references used (they are unclear and very few).

A better connection between IIoT and blockchain technology needs to be provided.

According to the methodology, not all the reader can be aware about game theory, so you need to provide a methodological section where you explain the main features and pillar of the Game based theory.

As for the future contributions and implications, I suggest to highlight the potential role of sustainability and digital transition issues as a future research direction for the topic investigated. I suggest improving the discussion of findings. With regard to the context of investigation, could it affect the results? Why? Additional information on how the evaluation is generated should be included to justify the significance of the results and clarify the effectiveness and rationality of the proposed approach. Moreover, the author/s should describe more in details if and how their application converges or diverges from other approaches? The authors should improve the theoretical justification according to previous methodological contributions. Finally, proofreading in different sections is needed.

Good luck!

Author Response

We appreciate the time and effort that you have dedicated to providing your valuable feedback on our manuscript. We are grateful for the insightful comments on our paper. Furthermore, we have been able to incorporate changes to reflect most of the suggestions provided. We have highlighted the changes within the manuscript. Here is a point-by-point response to the comments and concerns.

Comment 1: The authors should improve the theoretical justification of the study clarifying the references used (they are unclear and very few).

Response: Thank you for your comment. We modified references in introduction and state of art, changes highlighted in the attached file.

Comment 2: A better connection between IIoT and blockchain technology needs to be provided.

Response: We added more explanation in lines 53-58.

Comment 3: According to the methodology, not all the reader can be aware about game theory, so you need to provide a methodological section where you explain the main features and pillar of the Game based theory.

Response: we added subsection 4.1 for provide more explanation.

Comment 4: As for the future contributions and implications, I suggest to highlight the potential role of sustainability and digital transition issues as a future research direction for the topic investigated. I suggest improving the discussion of findings.

Response: we added more explanation regarding it in conclusion. 

Comment 5: Moreover, the author/s should describe more in details if and how their application converges or diverges from other approaches? The authors should improve the theoretical justification according to previous methodological contributions.

Response: we tried to add more detail in methodology part and clarify discussion section. 

Reviewer 4 Report

The paper discusses an industrial blockchain system architecture designed for multi-factory environment including blockchain integration supporting leverages of taxes. The paper is well organized and described.

Strengths: subnetworks and architectures improving cybersecurity.

Points of weakness: explanation and details about gaming approach;

A minor revision is required.

Actions to do:

According to the weaknesses, I suggest to improve the paper by answering to these points:

1-     Please provide more information/explanations (in particular about Fig. 5) about gaming approach and its application in the proposed architectures;  

2-     A ‘Discussion’ section should added by summarizing with a tables all the analyzed issues;

3-     More references should be added in the introduction section about Artificial intelligence AI  improving cybersecurity by increasing the perspectives of blockchain management systems, and  process mining approaches integrating blockchain in industry 4.0 applications , such as:

-        https://doi.org/10.3390/ai1010005 

-        DOI: 10.1109/ACCESS.2020.3034399

-        https://doi.org/10.3390/electronics11020198

-        https://doi.org/10.3390/s22228677

-        https://doi.org/10.3390/electronics11142128

4-     Conclusions should be improved. 

Minor remarks:

Please add more comment in all figure captions.

Author Response

We appreciate the time and effort that you have dedicated to providing your valuable feedback on our manuscript. We are grateful for the insightful comments on our paper. Furthermore, we have been able to incorporate changes to reflect most of the suggestions provided. We have highlighted the changes within the manuscript. Here is a point-by-point response to the comments and concerns.

Comment 1: Please provide more information/explanations (in particular about Fig. 5) about gaming approach and its application in the proposed architectures;

Response: Thank you for your comment. We added more explanation in lines 355-371.

Comment 2: A ‘Discussion’ section should added by summarizing with a tables all the analyzed issues;

Response: We added Table 4.

Comment 3: More references should be added in the introduction section about Artificial intelligence AI  improving cybersecurity by increasing the perspectives of blockchain management systems, and  process mining approaches integrating blockchain in industry 4.0 applications

Response: we added more explanation with mentioned references in lines 31-41.

Comment 4: Conclusions should be improved.

Response: we changed the conclusion in case of improving it.

Round 2

Reviewer 3 Report

Good luck.